# Evaluation of Oral Administration of *Lactobacillus plantarum* CAM6 Strain as an Alternative to Antibiotics in Weaned Pigs

**DOI:** 10.3390/ani10071218

**Published:** 2020-07-17

**Authors:** César Betancur, Yordan Martínez, Ruben Merino-Guzman, Xochitl Hernandez-Velasco, Rogel Castillo, Roman Rodríguez, Guillermo Tellez-Isaias

**Affiliations:** 1Departamento de Ciencias Pecuarias, Facultad de Medicina Veterinaria y Zootecnia, Universidad de Córdoba, Montería 230002, Colombia; cesarbetancur123@gmail.com; 2Science and Agricultural Production Department, Zamorano University, Francisco Morazán P.O. Box 93, Honduras; rcastillo@zamorano.edu; 3Facultad de Medicina Veterinaria y Zootecnia, Universidad Nacional Autónoma de México, Mexico City 04510, Mexico; onirem@unam.mx (R.M.-G.); xochitl_h@yahoo.com (X.H.-V.); 4Centro de Estudios de Producción Animal, Facultad de Ciencias Agropecuarias, Universidad de Granma, Bayamo 85149, Granma, Cuba; rrodriguezb@udg.co.cu; 5Department of Poultry Science, University of Arkansas, Fayetteville, AR 72701, USA; gtellez@uark.edu

**Keywords:** probiotic, weaned piglet, productivity, IgA, health status

## Abstract

**Simple Summary:**

Currently, due to intensive production, pigs are weaned early, which generates stress due to separation from the sow, metabolic disorders, and decreased productive performance. Thus, sub-therapeutic antibiotics have been used to alleviate these detrimental effects; however, it has been shown that these practices create microbial resistance and cross-resistance to other microorganisms. Although the European Union banned antibiotic growth promoters (AGP), many countries in the world still use them widely. In the present study, oral administration of *Lactobacillus plantarum* CAM6 in weaned pigs improved productive performance from the second experimental week and increased total serum levels of IgA without causing adverse effects on health indicators and acid-base balance. These results suggest this probiotic bacterium can be used as an alternative to antibiotics in weaned pigs.

**Abstract:**

The objective was to evaluate the effect of oral administration of *Lactobacillus*
*plantarum* CAM6 strain as an alternative to antibiotics in weaned pigs on productive parameters, blood biochemical profile, and IgA serum levels. Thirty-six 21-day-old weaned piglets were randomly assigned to three groups with three replicates of four piglets each. Treatments consisted of a basal diet (BD; T0) without probiotics or antibiotics; BD + antibiotics and the same basal diet used in T0 plus oral administration of 5 mL × 10^9^ CFU/mL of *L. plantarum* CAM-6 (T2). During the study (21 to 49 days of age) T2 obtained a similar live weight, weight gain, and feed conversion ratio when compared to the T1. Both treatments were better in these variables compared to T0 (*p* ≤ 0.05). Furthermore, T2 increased serum IgA levels (*p* ≤ 0.05). Additionally, hematological parameters and acid-base balance remained similar in all groups. However, significant reductions in the mean corpuscular hemoglobin concentration, platelets, and metabolic hydrogen ions were observed in T1 (*p* ≤ 0.05). The results of this study suggest that supplementation with *L. plantarum* CAM6 can be an alternative to antibiotics. Studies to evaluate its efficacy under commercial conditions and water administration require further evaluation.

## 1. Introduction

Weaning is one of the most stressful events in pig production [1]. Early weaned piglets face several nutritional and psychosocial stressors, including the transition to a solid diet and the abrupt separation of sows and piglets [2]. These events coincide with an immature immune and digestive system [3], which has consequences, such as the reduced activity of digestive enzymes, changes in intestinal morphology, and reduced nutrient digestion in the small intestine [4].

Furthermore, newly weaned pigs are immunodeficient [5]. Their intestinal immune system is fully mature at only 5 to 6 weeks of age [6], and the production of immunoglobulin A (IgA) antibodies start at four weeks of age [7]. This system is particularly crucial in mounting an effective immune response to protect the mucosa against antigens and allow the crosstalk between the immune systems of sows and piglets [8].

The intestinal morphology and immunodeficiency changes caused by early weaning allow for the growth of pathogens [9] and increase the need for antibiotics [10], which may lead to the development of resistant strains in the intestinal microbiota of humans and animals [11]. Probiotics can be used as alternatives to antibiotics. They do not cause side effects and increase animal production by improving digestibility, weight gain, feed conversion ratio [12], and physiological indicators of stress in weaned pigs [13].

Previous reports show that probiotics have a wide range of beneficial effects, including strengthening the function of the intestinal barrier [2], the inhibition of pathogenic bacteria [12], and immune development [6]. Among probiotics, *Lactobacillus* is the bacterial genus most commonly used in swine production [14], as weaning decreases the abundance of *Lactobacillus* in the intestine [15]. Moreover, it has been shown that *Lactobacillus plantarum* PFM 105 may improve the gut health of weaned pigs by regulating their intestinal microbiota [16]. For this study, we hypothesized that this probiotic strain could stimulate the immune system and increase the productivity of growing piglets. The objective of this study was to supplement the diet of weaned pigs with either a bacterial strain (*L. plantarum* CAM6) isolated from creole pigs (Zungo pelado), or antibiotics and determine their efficacy in improving productive parameters, blood biochemical profile and IgA serum levels. 

## 2. Materials and Methods 

The study was carried out in accordance with the Colombian guidelines for animal welfare, and the experiment was approved by the Animal Care and Use Committee of Córdoba University and Research (Resolution No. 1 of 26 January 2016).

### 2.1. Study Site

The research was carried out in the experimental pig farm of the University of Córdoba, Berástegui headquarters (Latitude 7°23′ 9°26′ north latitude and 74°52′ 76°3′ west longitude meridian of Greenwich and 30 meters above sea level), Córdoba, Colombia. The climate of the study area is classified according to the Holdridge classification as tropical wet and dry with two well-defined seasons, rainy and dry, an average annual temperature of 28 °C, relative humidity of 82%, and an average annual rainfall of 1400 mm [17]. 

Piglets born and weaned at the same time (21-days-old) from 20 healthy Yorkshire × Landrace (three farrowing’s) sows located in the farrowing area were randomly selected. Farrowing’s were synchronized, and no abnormalities were found during pregnancy and lactation. The experimental area where the pigs were located is built with a concrete floor and precast concrete blocks for the corral fronts and sides. The pens are 0.6 m high, providing ample space for natural ventilation.

### 2.2. Probiotic Preparation and Treatments

The *L. plantarum* CAM6 strain (GenBank accession number MK523644.1) was isolated from Creole pigs (Zungo pelado) from the north coast of Colombia. This bacterial strain was the most effective for in vitro probiotic tests. It demonstrated resistance to low pH (3.0), high bile salt (0.3%) and NaCl concentrations, high temperatures, and antibiotics, as well as antagonistic activity against bacterial pathogens [18]. 

This strain was inoculated in pineapple, banana, and papaya peel juice. Growth kinetics were performed to determine the most appropriate fruit peel concentration and optimize the substrate to the inoculum ratio and pH. The strain of *Lactobacillus plantarum* CAM-6 was inoculated at 10% (*v*/*v*) in an Erlenmeyer flask containing the juice at different concentrations. The fermentation was realized at environmental temperature (30 ± 2 °C), with constant shaking at 100 rpm in an orbital shaker (SK-o330-Pro LB PRO) for 24 h. The best medium consisted of 40% fruit peel juice and 60% water. The optimal substrate to the inoculum ratio was 6.81, and the best pH was 5.29. Under these conditions, a bacterial density of 10^9^ CFU mL^−1^ was obtained, and this concentration was used as a probiotic treatment. Treatments consisted of a basal diet (BD; T0) without probiotics or antibiotics; BD + 350 mg/kg colistin sulfate (20% of the active compound) and the same basal diet used in T0 plus oral administration of 5 mL × 10^9^ CFU/mL *L. plantarum* CAM-6 (T2). The ingredients and nutritional contributions are shown in Table 1 [19]. 

### 2.3. Animals and Experimental Conditions

The trial (28 days) used 36 piglets weaned at 21 days of age that were earmarked and randomly divided into three experimental groups under the same production conditions, with three replicates and four pigs per pen (3.60 × 1.90 each), where each pen was an experimental unit. All animals were dewormed and vaccinated against classical swine fever at 42 days of age. In each pen, feed was supplied ad libitum on two frequencies (8:00 a.m. and 3:00 p.m.) in linear canoe-type feeders. The water was supplied at will in metal nipple-type waterers.

The temperature of each pen was maintained at 26 ± 3 °C. One hour before the feed, an oral suspension with *L. plantarum* CAM-6 was provided. Piglets were orally gavaged with 5 mL of the probiotic using a syringe to ensure that all pigs received a similar volume of the bio preparation. The piglets were provided with the probiotic from day 21 to day 49 post-weaning.

### 2.4. Production Parameters

During the study, the live weights of the pigs were determined weekly from 21 to 49 days (4 weeks) using a scale with a precision ±1 g (Metter Toledo, digital Scale, Columbus, Ohio, USA), at the same time of day and before feeding. Feed intake was determined daily, and it was accumulated by the experimental week. The difference between feed supply and rejection was considered. The feed conversion rate was calculated as the amount of ingested feed required to gain 1 kg of LW. The average daily LW gain was determined from the difference between the final and the initial LW during the study.

### 2.5. Hematological Profile

On day 49 of age, eight pigs from each treatment group were randomly selected and 5 mL of blood was collected from their jugular veins with 21 G needles (Covidien, Mansfield, MA, USA) (1.5 inches) under sterile conditions. Blood plasma was collected in Vacutainer tubes with lilac caps (BD Vacutainer; BD, Franklin Lakes, NJ, USA) containing an anticoagulant agent (200 µL sodium heparin). Blood serum was collected in Vacutainer tubes (with red caps) without an anticoagulant agent and was centrifuged at 3000× *g* for 15 min (Eppendorf centrifuge AG, New York, NY, USA) at room temperature.

The following hematological variables were evaluated in a semi-automatic analyzer (Horiba ABX Micros ESV 60^®^; Paris, France): White blood cells (WBC); red blood cells (RBC); hemoglobin; hematocrit; mean corpuscular volume (MCV); mean corpuscular hemoglobin concentration (MCHC); mean cell volume (MCV); mean platelet volume. All samples were processed in the Veterinary Clinical Laboratory of the School of Veterinary Medicine and Zootechnics of the University of Córdoba, Colombia. 

### 2.6. Biochemical Parameters

In addition to the same selected animals, the acid-base balance (ABB) was measured in plasma, and beta-hydroxybutyrate (BHB) and IgA were measured in serum. The ABB was performed in a blood gas analyzer, RapidLab 348 (Siemens-Bayer, Berlin, Germany), with selective ion electrodes, while pH and pCO_2_ were measured using test strips. HCO_3_^−^ and the excess of bases were calculated automatically by the equipment (RapidLab 348; Simens-Bayer, Germany). Metabolic hydrogen ions (M*H) was measured as total hydrogen ions minus the H* derived from CO_2_ (HCO_3_^−^) in the blood. The commercial kit, Rambut^®^
d-3-Hydroxybutyrate–RB 1007 (Randox Laboratories, Crumlin, County Atrium, UK), was used to evaluate BHB using an ultraviolet spectrophotometer (Metrolab 1600, New York, NY, USA). Total IgA serum concentration was measured using a commercial ELISA kit (Cat. No. E100-102; Bethyl Laboratories Inc., Montgomery, TX, USA), as previously described [20].

### 2.7. Statistical Analysis

All data were subjected to analysis of variance (ANOVA) as a completely randomized design. For the normality of the data, the Kolmogorov–Smirnov test was used, and for the uniformity of the variance, Bartlett’s test was used before carrying out the ANOVA. For the LW and ADG, each pig is considered as a replicate and, for the FI and FCR, each pen was considered as an experimental unit. Treatment means were partitioned using Duncan’s multiple range test at *p* < 0.05 indicating statistical significance using the statistical software SPSS version 23.0 (SPSS Inc., IBM Corporation, New York, NY, USA).

## 3. Results

Table 2 shows the results of the effect of *Lactobacillus plantarum* CAM6 probiotic on the growth performance of weaned pigs. There were no significant differences in the LW of animals aged 21 to 28 days among the groups (*p* < 0.05). However, in the 35–49-day period, live weight increased with T1 and T2 (*p ≤* 0.05). Similarly, the probiotic and antibiotics groups improved (*p ≤* 0.05) the average daily gain in the overall period of the piglets compared to the control group, with a decrease in the feed conversion ratio (21–49 days) and without affecting feed intake (21–49 days). The performance was similar between probiotic and antibiotic supplemented groups.

The effects of *Lactobacillus plantarum* CAM6 probiotic on the hematological parameters of weaned pigs are summarized in Table 3. Statistical differences were observed among treatments (*p* ≤ 0.05) for MCHC and platelets in both cases with the lowest antibiotic group values. The other blood parameters did not show significant differences among treatments (*p* < 0.05).

Table 4 shows the results of the effect of *Lactobacillus plantarum* CAM6 probiotic on the acid-base balance of weaned pigs. The only indicator that changed statistically (*p* ≤ 0.05) due to the effect of the experimental treatments was M*H, with a decrease in T1 compared to T0.

The results of the effect of oral administration with *Lactobacillus plantarum* CAM6 as a probiotic candidate in vivo on the concentration of serum BHB and total IgA in weaned pigs are summarized in Table 5. The BHB concentration was statistically higher (*p* ≤ 0.05) in T0 and T2. Furthermore, it is observed that T2 significantly increased IgA) (*p* ≤ 0.05) in relation to T0.

## 4. Discussion

The viability (100%; data not shown) was excellent for all treatments, which indicated the innocuousness the probiotic product used from 21 to 49 days. Note that no symptoms, signs, or pathologies associated with any disease were found. Other authors who used probiotics in pigs found similar results in viability [21,22,23], demonstrating that this lactic acid bacteria strain (*Lactobacillus plantarum* CAM-6) is safe for pig production. One of the goals of this study was to determine whether oral administration with *Lactobacillus plantarum* CAM-6 in pigs would influence growth performance, as the primary justification for use. 

Thus, the most stressful stage during animal development is during the first two weeks post-weaning and is characterized by the transition from a liquid (milk) to a solid diet [21], which may cause digestive disturbances, reduced feed intake, and stunted growth in newly weaned piglets [22]. Enzyme maturation begins at this stage. The presence of *Lactobacillus* accelerated this process, resulting in an increase in the LW gain of animals in the T2 groups compared to control, as shown in Table 2. In this sense, Jensen et al. [23] observed that the enzyme system was fully developed in piglets in their eighth week of life. The LW gain of animals in the T1 and T2 groups could be due to the positive effect of probiotics on the intestinal microbiota. It has been reported that probiotics improve animal growth [16,24]. 

Besides, the improved LW gain observed upon the feeding of antibiotics may be associated with the regulation of intestinal microbiota, which improves the reuptake of nutrients [25]. The mean daily LW gain was significantly higher on the overall period (21–49 days), which was similar between probiotic and antibiotic supplemented groups. This result could be associated with the improved digestive efficiency favored by probiotics and antibiotics, which reduced the adverse effects of weaning. Vrotniakiene and Jatkauskas [26] reported that the daily LW gain increased when pigs were supplemented with probiotics. It is known that lactobacilli release enzymes that improve digestive capacity, increase intestinal absorption, and lead to an increase in daily LW gain and the growth rate of pigs supplemented with probiotics [2]. Similarly, the improved growth performance in T1 is consistent with the results of a previous study. A therapeutic dose of different antibiotics supplemented to the diet improved the average growth performance of the animals during the fourth-week post-weaning [27].

Furthermore, these data show that the dietary use of the antibiotic and *Lactobacillus plantarum* CAM6 does not statistically modify feed intake. Feed conversion ratio during the post-weaning of pigs (21–49 days) was significantly lower in the T2 group compared to the T1 and T0 groups, as shown in Table 2. Probiotics improve animal growth by increasing the height of the jejunum villi [28,29], thereby causing an increase in nutrient absorption, which may explain the best-feed conversion ratio observed in the T2 group. These results demonstrate the effect of supplements on the digestive physiology of animals, evidenced by the increase in the LW gain and the decrease in the feed conversion rate water absorption capacity of the intervention groups [16].

Hematological parameters are currently used as indicators of human and animal health. Variations in these indicators may reflect bacterial, viral, parasitic, or fungal infections, as well as problems with blood poisoning, dehydration, or clotting [10]. In general, it is necessary to determine if a new product, such as probiotics, can cause changes in these blood parameters. Some compounds’ functional foods can induce changes in polymorphonuclear leukocytes (neutrophils and eosinophils), mainly activating the immune system to eliminate exogenous material and/or possible toxic and allergenic compounds [30]. There were no significant differences among the cell counts, hemoglobin, hematocrit, average cell volume, and average corpuscular hemoglobin. Those parameters were within normal limits [30] and are consistent with previous studies [31,32].

Interestingly, a significant reduction in the mean corpuscular hemoglobin concentration and platelets were observed in the pigs that received the feedstuff with antibiotics when compared with the other two groups, as shown in Table 2. Intriguingly, during the experimental stage, the diarrheal syndrome did not appear, which shows that the experimental conditions were adequate. Furthermore, the hematocrit was maintained without significant changes between the experimental treatments. Diarrheal syndrome causes the loss of electrolytes and water, which creates a hemoconcentration in the blood and an increase in the hematocrit [33]. Likewise, there were no significant differences between the ABB between groups, as these values were within the reference limits [30]. However, there was a significant reduction in the metabolic hydrogen ions between the concentrate without probiotics or antibiotics group and the group that received concentrate with antibiotics. Medications, including antibiotics, may cause hypokalemia, which may lead to metabolic alkalosis [34]. Therefore, the ABB is a useful physiological stress indicator in farm animals. 

A significant reduction in BHB concentration was observed in the group that received the feedstuff with antibiotics when compared to the control without probiotics or antibiotics or the feedstuff with probiotics. After weaning, different stressors cause the release of hormones and cytokines, resulting in reduced feed consumption [3]. The ketone body levels may increase, owing to reduced caloric intake and anorexia. The results of a study showed that newly weaned piglets with anorexia had increased serum levels of BHB [35]. Animals from the T1 group had lower BHB levels, which could be due to the beneficial effect of antibiotics in the digestive system. Antibiotics can reduce the BHB concentration, as evidenced by the beneficial effects of some medications on digestive metabolism, as shown in Table 4. The metabolome analysis after antibiotic treatment showed an improvement in several energy metabolism pathways in pigs [36]. However, the probiotic group had a significant increase in total serum levels of IgA when compared with the group that received the feedstuff without probiotics or antibiotics. These results suggest an improvement in the humoral immune response of weaned piglets that received the supplementation of *L. plantarum* CAM6 compared to weaned piglets given an antibiotic treatment, as shown in Table 4. Similar results were found by Wang et al. [21], wherein dietary supplementation with *L. plantarum* ACCC 11016 plus fructooligosaccharides increased serum IgA concentrations compared to the use of the antibiotic aureomycin. Immunoglobulin concentration is an essential criterion for assessing the humoral immune response in animals [37]. Liao and Nyachoti [38] have shown that probiotics improve immunity by increasing the intestinal absorption of nutrients and immunoglobulins. In summary, the results of the present study suggest that the oral administration of *L. plantarum* CAM6 can be an alternative to antibiotic growth promoters in weaned pigs. Studies to evaluate its efficacy under commercial conditions and water administration require further investigation.

## 5. Conclusions

The results of the present study suggest that supplementation with the *L. plantarum* CAM6 improves the productive parameters and the concentration of IgA levels without affecting health indicators in weaned pigs.

## Figures and Tables

**Table 1 animals-10-01218-t001:** Ingredients and nutritional contributions of the pig diet (21–49 days).

Ingredients, %	Percent
Cornmeal	58.40
Soybean meal	29.78
Wheat bran	3.52
Vegetable oil	3.00
Methionine	0.22
Starter nucleus ^1^	2.50
Monocalcium phosphate	0.90
Calcium carbonate	1.28
Common salt	0.40
Nutritional contributions, %	
Crude protein	19.0
Lysine	1.06
Methionine + cystine	0.72
Tryptophan	0.22
Calcium	0.76
Total phosphorus	0.62
Crude fat	6.04
Crude fiber	2.73
Metabolizable energy (kcal kg^−1^)	3285

^1^ The vitamin and mineral premix provided per kilogram of diet: 20,000 IU vitamin A; 4000 IU vitamin D3; 80 IU vitamin E; 16 mg vitamin K; 4 mg thiamine; 20 mg riboflavin; 6 mg pyridoxine; 0.08 mg vitamin B12; 120 mg niacin; 50 mg Ca-pantothenate; 2 mg folic acid; 0.08 mg biotin; 15 mg Cu (as copper sulfate); 56 mg Zn (as zinc oxide); 73 mg Mn (as manganese oxide); 0.3 mg I (as potassium iodine); 0.5 mg Co; 0.4 mg Se.

**Table 2 animals-10-01218-t002:** Effect of oral administration with *Lactobacillus plantarum* CAM6 on the growth performance of weaned pigs.

Days of Age	Treatment			SEM ±	*p*
	T0	T1	T2		
Live weight, kg					
21	5.69	5.95	5.94	0.284	0.820
28	6.94	7.49	7.68	0.370	0.714
35	8.25 ^b^	9.36 ^a^	9.38 ^a^	0.235	0.038
42	9.79 ^b^	10.98 ^a^	10.80 ^a^	0.271	0.041
49	11.50 ^b^	12.61 ^a^	12.88 ^a^	0.423	0.018
Average daily gain, g					
21–49	209.92 ^b^	237.92 ^a^	247.32 ^a^	9.083	0.043
Feed intake, g/day					
21–49	416.55	421.48	419.62	1.893	0.873
Feed conversion ratio, kg/kg					
21–49	2.05 ^a^	1.89 ^a, b^	1.78 ^b^	0.079	0.045

^a, b^ Means with different superscripts within a row differ significantly (*p ≤* 0.05). T0: feedstuff without probiotics or antibiotics; T1: feedstuff with antibiotics; T2: feedstuff with 10^9^ CFU *L. plantarum* CAM6/mL. *n* = 12 pigs per treatment for live weight and weight gain; *n* = 3 pens per treatment to determine the feed intake and feed conversion. SEM: standard error of the mean.

**Table 3 animals-10-01218-t003:** Effect of oral administration with *Lactobacillus plantarum* CAM6 on the hematological parameters of weaned pigs.

Indicators	Treatment			SEM	*p*
	T0	T1	T2		
Leukocytes, 10^3^/mm^3^	17.90	16.25	18.88	1.329	0.856
Lymphocytes, 10^3^/mm^3^	4.43	4.93	5.98	0.564	0.315
Monocytes, 10^3^/mm^3^	1.13	1.25	1.28	0.127	0.847
Granulocytes, 10^3^/mm^3^	12.35	10.07	11.62	1.897	0.811
Eosinophils, 10^3^/mm^3^	0.43	0.61	0.44	0.098	0.789
Erythrocytes, 10^6^/mm^3^	6.63	7.15	6.89	0.330	0.748
Hb, g/dL	10.75	11.17	10.73	0.447	0.066
Hto, %	33.30	35.05	32.40	1.820	0.762
ACV, µm^3^	48.50	49.00	47.25	1.358	0.829
ACH, pg	16.25	15.70	15.55	0.277	0.072
MCHC, g/dL	33.63 ^a^	31.98 ^b^	33.10 ^a^	0.253	0.001
Platelets, 10^3^/mm^3^	1262.75 ^a^	760.50 ^b^	1120.75 ^a, b^	36.281	0.043
APV, µm^3^	13.20	13.57	14.27	1.035	0.745

^a, b^ Means with different superscripts within a row differ significantly (*p* ≤ 0.05). T0: feedstuff without probiotics or antibiotics; T1: feedstuff with antibiotics; T2: feedstuff with 10^9^ CFU *L. plantarum* CAM6/mL. Hb: hemoglobin; Hto: hematocrit; ACV: average cell volume; ACH: average corpuscular hemoglobin; MCHC: mean corpuscular hemoglobin concentration, APV: average platelet volume. *n* = 8 pigs per treatment. SEM: standard error of the mean.

**Table 4 animals-10-01218-t004:** Effect of oral administration with *Lactobacillus plantarum* CAM6 on the acid-base balance of weaned pigs.

Indicators	Treatment			SEM	*p*
	T0	T1	T2		
pH	7.29	7.38	7.31	0.041	0.262
pCO_2_ (mm/Hg)	56.40	49.18	55.45	4.867	0.545
HCO_3_^−^ (mEq/L)	25.78	27.03	26.93	0.690	0.480
EB (mEq/L)	−1.37	1.55	0.05	1.087	0.218
H*M	0.875 ^a^	−5.26 ^b^	−1.47 ^a, b^	1.797	0.010

^a, b^ Means with different superscripts within a row differ significantly (*p* ≤ 0.05). T0: feedstuff without probiotics or antibiotics; T1: feedstuff with antibiotics; T2: feedstuff with 10^9^ CFU of *L. plantarum* CAM6/mL. pCO_2_: CO_2_ pressure; EB: excess bases; HCO_3_^−^: bicarbonate; M*H: metabolic hydrogen ions. *n* = 8 pigs per treatment. SEM: standard error of the mean.

**Table 5 animals-10-01218-t005:** Effect of oral administration with *Lactobacillus plantarum* CAM6 on serum d-β-hydrobutyrate and immunoglobulin A concentration in weaned pigs.

Indicators	Treatment			SEM	*p*
	T0	T1	T2		
BHB (mg/dL)	1.25 ^a^	0.68 ^b^	1.15 ^a^	0.145	0.017
IgA (ng/mL)	5.728 ^b^	6.597 ^a, b^	7.217 ^a^	0.712	0.006

^a, b^ Means with different superscripts within a row differ significantly (*p* ≤ 0.05). T0: feedstuff without probiotics or antibiotics; T1: feedstuff with antibiotics; T2: feedstuff with 10^9^ CFU of *L. plantarum* CAM6/mL. BHB: d-β-hydrobutyrate; IgA: Immunoglobulin A. *n* = 8 pigs per treatment. SEM: standard error of the mean.

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
