# Peer review of "Evaluation of Oral Administration of Lactobacillus plantarum CAM6 Strain as an Alternative to Antibiotics in Weaned Pigs"

_animals, 2020, doi:10.3390/ani10071218_

Round 1

Reviewer 1 Report

The article "Oral administration with Lactobacillus 2 plantarum CAM6 Strain improves productivity and 3 immune functions in weaned pigs" is suitable for the purpose of the journal and, moreover, is characterized by a high level of novelty. The experimental design is appropriate. The article is thus published after a minor revision.
1) Chemical and hematological analyzes were performed on fewer pigs than the pigs used to measure in vivo performance, so, the exact number of pigs tested for each analysis should be better specified.
2) In addition, the statistical analysis model should be more detailed by describing the fixed factors used.

Author Response

Response to Reviewer 1 Comments

The article "Oral administration with Lactobacillus 2 plantarum CAM6 Strain improves productivity and 3 immune functions in weaned pigs" is suitable for the purpose of the journal and, moreover, is characterized by a high level of novelty. The experimental design is appropriate. The article is thus published after a minor revision.

Point 1. Chemical and hematological analyzes were performed on fewer pigs than the pigs used to measure in vivo performance, so, the exact number of pigs tested for each analysis should be better specified.

Response 1. Done. The number of animals was indicated for each analysis.

Point 2. In addition, the statistical analysis model should be more detailed by describing the fixed factors used.

Response 2. We indicated “The data were processed using one-way analysis of variance (ANOVA), for the normality of the data the Kolmogorov Smirnov test was used and for the uniformity of the variance Bartlett's test was used, before carrying out the ANOVA. The Duncan’s test was used to determine the differences between means, according to the statistical software SPSS version 23 (SPSS. IBM -International Business Machine, Nueva York. USA) P values <0.05 were considered statistically significant”

Reviewer 2 Report

The study is interesting and in line with the need to replace antibiotics with natural growth-promoting substances

Author Response

Response to Reviewer 2 Comments

Point 1. The study is interesting and in line with the need to replace antibiotics with natural growth-promoting substances

Response 1. Thank very much for your comments.

Reviewer 3 Report

Materials and methods

2.2

line 84-87: add literature

line 90: fruit peels or fruit peel juice?

Line 9: How was added the liquid supplement to the dry feed to ensure a homogeneous distribution?

Line 97:   at which age of the piglets started the trial? “The piglets were included in the study from day 21 to day 49 post-97 weaning “means that the trial started 21 days after weaning, not that pigs were 21 days old at the start of the trial.

2.4

line 111: “the initial and final LW of each animal at days 21 and 49 of age” means that the piglets were 21 days old at the start of the trial.

Line 111-112: the pigs were weighed at the start (21 days) and at the end (49 days) of the trial or every week as reported in table 2?

TABLE 1: the diet formulated with 58.4 of corn meal (lysine 2.7 g/kg) , 29.78 g/kg of soybean meal (if 48% crude protein lysine 30 g/kg) and 3.52 of wheat bran (lysine 6 g/kg ) can contain no more than 1.07 of lysine. The starter nucleus contained synthetic L- lysine?

More information is needed about starter nucleus composition and carrier.

2.5

The hematological profile and biochemical parameters were measured on 5 piglets and only at the end of the trial. How can you be sure that no differences may be present also at the start?

2.7

At which piglets age was performed the vaccination?

Five piglets are too few for an immunological essay, in particular if the piglets were positive before the vaccination. Why are the piglets positive before the vaccination? Were the sows positive or vaccinated? Which kind of ELISA test was used? In table 5 how is expressed the serological response?

Author Response

Response to Reviewer 3 Comments

Thanks for your comments

Materials and methods

2.2

Point 1. Line 84-87: add literature

Response 1. Done

Point 2. line 90: fruit peels or fruit peel juice?

Response 2. Done. We added the word "juice"

Point 3. Line 9. How was added the liquid supplement to the dry feed to ensure a homogeneous distribution?

Response 3. We added an explanation Page 3, line 116:

Point 4. Line 97:   at which age of the piglets started the trial? “The piglets were included in the study from day 21 to day 49 post-97 weaning “means that the trial started 21 days after weaning, not that pigs were 21 days old at the start of the trial.

Response 4. The initial LW was at day 21 and the final LW was at 49 days of age.

Point 5. line 111: “the initial and final LW of each animal at days 21 and 49 of age” means that the piglets were 21 days old at the start of the trial.

Response 5. The pigs were weaned at 21 days and this moment began the trial.

Point 6: Line 111-112: the pigs were weighed at the start (21 days) and at the end (49 days) of the trial or every week as reported in table 2?

Response 6. The pigs were weighed every week.

Point 7. TABLE 1: the diet formulated with 58.4 of corn meal (lysine 2.7 g/kg) , 29.78 g/kg of soybean meal (if 48% crude protein lysine 30 g/kg) and 3.52 of wheat bran (lysine 6 g/kg ) can contain no more than 1.07 of lysine. The starter nucleus contained synthetic L- lysine?

More information is needed about starter nucleus composition and carrier.

Response 7. There was an error in the contributions of lysine, we bought the ready-made feed, because we did not modify its ingredients. The value was rectified.

Point 8. The hematological profile and biochemical parameters were measured on 5 piglets and only at the end of the trial. How can you be sure that no differences may be present also at the start?

Response 8. Actually, eight animals per treatment were used for hematological and biochemical indicators. These animals are apparently healthy, which means that the effect of the additives was the only one that could influence the variables.

Point 9. At which piglets age was performed the vaccination?

Five piglets are too few for an immunological essay, in particular if the piglets were positive before the vaccination. Why are the piglets positive before the vaccination? Were the sows positive or vaccinated? Which kind of ELISA test was used? In table 5 how is expressed the serological response?

Which kind of ELISA test was used

Response 9. You are right. This section was removed from Table 5 because the number of samples is very low.

Reviewer 4 Report

General comment

Weaned piglets are extremely sensitive to environmental condition, first of all nutrition. Feed compounds for weaning period must be formulated very precisely, including not only nutritional requirements of animals but also their health and physiological function of digestive tract. One of the method to increase digestibility of nutrients and improve gut health is the use of probiotics or/and prebiotics that have been considered as most important replacement for antibiotics. There are large number of different probiotic preparations in the market worldwide, and effectiveness of their use in production is diverse. The Author propose to use novel solution, bacterial strain isolated from creole pigs. The manuscript is written proper language, however, needs some minor language revision. Unfortunately, many imprecision and unclear information were found, especially in M&M, what makes it difficult to assess real value of research and paper.

Specific comments

Introduction

Introduction is relatively short but well written and informative section. However, there is a lack of important information about the mechanisms of probiotic action in intestine. The number of papers and information in this area is rather large. The Authors should also try to hypothesize why proposed bacterial strain will be better solution than others being presently used. This should be corrected and completed.   

M&M

Undoubtedly this section is the worst in whole manuscript. Many of information is unclear, many important data is missed or described imprecisely, and some information does not match with results. This make all the manuscript difficult to assess in terms of data reliability and correctness of conclusions. Whole M&M section needs deep, substantial improvement.

Line 77-81: The description of farm should be included (number of sows in base herd, production cycle, number of sows weaned per cycle, ventilation type etc.)

Line 85-86: What is the difference between salt and NaCl?

Line 88-92: Was the probiotic prepared by the Authors, or purchased commercially? If prepared by the Authors, the description needs some additional information like the time and temperature of incubation (including the type of trademark of incubator), and the method of bacterial density assessment and strain recognition.    

Line 93-94: Could the authors expand on the randomization procedures used to allocate pigs to treatment groups? The description is unclear. What does it mean 3 replicates? Was the replicates provided the same time, or every replicate in different term?

Line 95: What was the concentration of colistin in full diet? It is not enough to show the weight of preparation used, if nothing is known about concentration of active substance in the product.

Line 95-96: Was the probiotic in liquid or dried form? It is not clear from description, but the information that it was added 10 mL suggest liquid form. It is also not clear if the addition was per 1 kg of diet (one can guess this because of antibiotic addition description which was per 1 kg, but M&M is not the place to guess, the information should be clear and complete).

Line 99-104 (table 1): The total volume of ingredients being listed in table is 100% (it seem to be diet T0). Including colistin in diet T1 or probiotic in diet T2 the Authors had to withdraw something, but they did not pointed what was it. One of the most problematic nutrient in weaning piglets is fat, because of low level and activity of bile acids and lipase. That is why the concentration of fat in diet should be no larger than 5-6%. The Authors did not give any information about the concentration of crude fat in diets. In turn, one of the most important regulator of gut function and health is crude fiber. There is also no information about this important nutrient in table 1. Taking into account little variety of compounds (the only cereal ingredient was corn), it can be predicted that the diets were reach in fat and poor in fiber, but this should be clearly descript.

Line 106-107: Space allowance in the pen seem to be very generous (1,71 sq m/pig). Could the Authors explain, why they decided to use only 4 pigs per so large pen. This is impossible that anybody  in commercial production will house weaning pigs with such a space allowance.  

Line 107-108: Could the Authors provide more detailed description of feeder type? The most important: Is it equipped with tray? The Authors claimed that feed was provided ad libitum, but was offered at 8.00 am and 3.00 pm. If it was ad libitum it does not matter when was offered. But if the feeder has no tray, it is impossible that it was ad libitum.

Line 111-112: The initial and final body weight were measured. It does not match with table 2 where changes in body weights were analysed in seven days interval. Could the Authors explain this discrepancy? Was 21 day the day of weaning?

Line 133-114: How actual intake was measured?

Line 118: why only 5 pigs per treatment were selected to hematological analyses. The analyses in semi-automatic analyzer is cheap procedure, and it is difficult to understand why did not the Authors analyze all pigs? Another matter is the number of 5 pigs. This number completely does not match to the number of pigs in groups. Much more sensible would be selection of 6 pigs (2 per pen). Selecting 5 pigs there is some possible configurations in 3 pens (2+2+1, or 3+1+1, or 4+1+0 etc.). The Authors have clarify this point.

Line 149: I am not sure that completely randomized design is the best solution in such a small number of experimental animals. In my opinion, the animals should be selected to treatment groups by drawing 3 similar animals from one litter and divide them into 3 groups, 1 animal per group. In such a situation the design is still randomized, but the division of pigs from litters is more valuable for results. I would like the Authors to descript more detailed the system of randomized division. How many litters were used? Did they use every piglet from litters, or were they selected (only large or only middle weight pigs). It is especially important according to data in table 2, where the mean initial body weight in T0 is insignificantly, but in my opinion relevantly lower than in groups T1 i T2 (this two groups are almost identical). This difference can be pivotal for the rest of results in equal level to experimental factors.

Nothing is known about the distribution of data. Did Authors analyze it? Was it parametric or non parametric? Which test was used? If the data was parametric the use of ANOVA is justified, but if it was non parametric the other tests should be used. All this doubts must be clarified.

One more important issue. The Authors did not provide any data on health status of pigs. In my opinion important and very useful would be the data about diarrhea occurrence. Looking at the table 2 data (average daily gains and feed conversion ratio) it seems to be clear that something was wrong in T0 group among day 42 and 49. So rapid decrease in gains must have a reason in health. However, in such a small group it also is possible, that the reason could be one sick pig. Such a pig should be excluded from analyses as outlier. That is why I would suggest to show SD or SEM for every group separately. Larger level of dispersion in group T0 would be important information. I would like the Authors also to clarify the n in consecutive analyses. I assume that weights and gains are individual, so n=12 per treatment. But it is not clear from the table. But what about feed intake and feed conversion ratio? I assume that n=3 (like the number of pens), but it is only assumption. This (n) should be clarified in every table, because there were important differences among analyses.

Results

Results are stated in 5 tables. In such a simple experimental design I would also calculate SEM separately for data inside groups (as I mentioned earlier). Sometimes such an analyses may show interesting, additional relations (e.g. if some parameters in one group are more dispersed than in the second one it is also important information). And last but not least, in every table n value for each group should be presented, because of imprecise description in M&M. The reader must know the number of animals in particular analyses to assess reliability and value of data.

Discussion

The most of this section seem to be continued Introduction and Results in a little more expanded manner, rather than actual discussion. The most important information are only rewritten from results without any trial to explain or interpret. It is easy to say that the growth of pigs from T0 group was lower, but much more important is to try to find answer why. Such information should be analyzed with the trial to interpret. As I said earlier, maybe the reason was 1 or 2 pigs which experienced diarrhea. Interesting but confusing result is CSF vaccination effect. It cannot be stated that reaction was increased in T2 group, because the level of antibodies was almost the same before and after treatment. Important question is why the level of antibodies was increased in T2 in comparison to T1 and T0 before vaccination? And why, this level decreased in T0 and T1 group after vaccination? It does not make a sense. And the Authors did not try to analyze and interpret this data in Discussion.

To summarize, the manuscript needs substantial improvement, and must be completed with many important data and information. Without this improvement, it is very difficult to assess the reliability of presented data. The most important seem to be clarification of randomized model of division. Taking into account the character of data and experimental design, the number of 12 animals per group for production analyses, and 5 animals for lab analyses seem to be not enough for the full paper. I suggest major revision, but also to rewrite the manuscript as short communication.

Author Response

Response to Reviewer 4 Comments

General comment

Weaned piglets are extremely sensitive to environmental condition, first of all nutrition. Feed compounds for weaning period must be formulated very precisely, including not only nutritional requirements of animals but also their health and physiological function of digestive tract. One of the method to increase digestibility of nutrients and improve gut health is the use of probiotics or/and prebiotics that have been considered as most important replacement for antibiotics. There are large number of different probiotic preparations in the market worldwide, and effectiveness of their use in production is diverse. The Author propose to use novel solution, bacterial strain isolated from creole pigs. The manuscript is written proper language, however, needs some minor language revision. Unfortunately, many imprecision and unclear information were found, especially in M&M, what makes it difficult to assess real value of research and paper.

Specific comments

Point 1. Introduction

Introduction is relatively short but well written and informative section. However, there is a lack of important information about the mechanisms of probiotic action in intestine. The number of papers and information in this area is rather large. The Authors should also try to hypothesize why proposed bacterial strain will be better solution than others being presently used. This should be corrected and completed.   

Response 1. Done. We write down some aspects of our strain and add more literatures.

Point 2. M&M  Undoubtedly this section is the worst in whole manuscript. Many of information is unclear, many important data is missed or described imprecisely, and some information does not match with results. This make all the manuscript difficult to assess in terms of data reliability and correctness of conclusions. Whole M&M section needs deep, substantial improvement.

Line 77-81: The description of farm should be included (number of sows in base herd, production cycle, number of sows weaned per cycle, ventilation type etc.)

Response 2. Done. We added more information in M&M considering your comments.

Point 3. Line 85-86: What is the difference between salt and NaCl?

Response 3. Line 85 Salt in bile salt is referred to bile acids with cations (principally with Na) for this reason is nameled bile salts (Gonzales-Gallego, 1995) and NaCl is a mineral mix used in the laboratory.

Point 4. Line 88-92: Was the probiotic prepared by the Authors, or purchased commercially? If prepared by the Authors, the description needs some additional information like the time and temperature of incubation (including the type of trademark of incubator), and the method of bacterial density assessment and strain recognition.    

Response 4. Done. “The Lactobacillus plantarum strain was inoculated at 10% (v / v) in each of the erlenmeyers. containing the juice at different concentrations.  The fermentation was carried out at room temperature (30 ± 2° C), with constant agitation at 100 rpm in an orbital shaker (SK- o330-Pro LB PRO) for 24 h”.

Point 5: Line 93-94: Could the authors expand on the randomization procedures used to allocate pigs to treatment groups? The description is unclear. What does it mean 3 replicates? Was the replicates provided the same time, or every replicate in different term?

Response 5. The piglets were housed on three pens, each pen measured 3.60 × 1.90 m (four animals per pen), where each pen constitute one experimental unit.

Point 6. Line 95: What was the concentration of colistin in full diet? It is not enough to show the weight of preparation used, if nothing is known about concentration of active substance in the product.

Response 6. The concentration of colistin was of 350 mg/kg of feed.

Point 7. Line 95-96: Was the probiotic in liquid or dried form? It is not clear from description, but the information that it was added 10 mL suggest liquid form. It is also not clear if the addition was per 1 kg of diet (one can guess this because of antibiotic addition description which was per 1 kg, but M&M is not the place to guess, the information should be clear and complete).

Response 7. The probiotic was administered in liquid form and 5 ml per kg of feed was added, as described in M&M.

Point 8. Line 99-104 (table 1): The total volume of ingredients being listed in table is 100% (it seem to be diet T0). Including colistin in diet T1 or probiotic in diet T2 the Authors had to withdraw something, but they did not pointed what was it. One of the most problematic nutrient in weaning piglets is fat, because of low level and activity of bile acids and lipase. That is why the concentration of fat in diet should be no larger than 5-6%. The Authors did not give any information about the concentration of crude fat in diets. In turn, one of the most important regulator of gut function and health is crude fiber. There is also no information about this important nutrient in table 1. Taking into account little variety of compounds (the only cereal ingredient was corn), it can be predicted that the diets were reach in fat and poor in fiber, but this should be clearly descript.

Response 8. We add more details in the diet provided. It is important noted that antibiotic and probiotic are used as additives

Point 9. Line 106-107: Space allowance in the pen seem to be very generous (1,71 sq m/pig). Could the Authors explain, why they decided to use only 4 pigs per so large pen. This is impossible that anybody in commercial production will house weaning pigs with such a space allowance.  

Response 9. From a total area of 6.84 it was divided into 4 sections, and in each part (1.71 m2) 4 pigs were housed per each section. A space of 1.71 m2 was left free for access to supply the probiotic. The stocking density was 0.42 m2 /piglet. The EU animal welfare regulations require a minimum space of 0.65 m2 / pig for a live weight of 100 kg.

Point 10. Line 107-108: Could the Authors provide more detailed description of feeder type? The most important: Is it equipped with tray? The Authors claimed that feed was provided ad libitum, but was offered at 8.00 am and 3.00 pm. If it was ad libitum it does not matter when was offered. But if the feeder has no tray, it is impossible that it was ad libitum.

Response 10. We supply the water ad libitum and the food was supplied 2 kg half in the morning and the other part in the afternoon.

Point 11. Line 111-112: The initial and final body weight were measured. It does not match with table 2 where changes in body weights were analysed in seven days interval. Could the Authors explain this discrepancy? Was 21 day the day of weaning?

Response 11. The pigs were weaned at 21 days and in this moment begin the trial. The initial LW was at day 21 and the final LW was at 49 days of age.

Point 12. Line 133-114: How actual intake was measured?

Response 12. By the offer-reject method

Point 13. Line 118: why only 5 pigs per treatment were selected to hematological analyses. The analyses in semi-automatic analyzer is cheap procedure, and it is difficult to understand why did not the Authors analyze all pigs? Another matter is the number of 5 pigs. This number completely does not match to the number of pigs in groups. Much more sensible would be selection of 6 pigs (2 per pen). Selecting 5 pigs there is some possible configurations in 3 pens (2+2+1, or 3+1+1, or 4+1+0 etc.). The Authors have clarify this point.

Response 13. This was a transcription error; 8 samples were used per treatment....

Point 14. Line 149: I am not sure that completely randomized design is the best solution in such a small number of experimental animals. In my opinion, the animals should be selected to treatment groups by drawing 3 similar animals from one litter and divide them into 3 groups, 1 animal per group. In such a situation the design is still randomized, but the division of pigs from litters is more valuable for results. I would like the Authors to descript more detailed the system of randomized division. How many litters were used? Did they use every piglet from litters, or were they selected (only large or only middle weight pigs). It is especially important according to data in table 2, where the mean initial body weight in T0 is insignificantly, but in my opinion relevantly lower than in groups T1 i T2 (this two groups are almost identical). This difference can be pivotal for the rest of results in equal level to experimental factors.

Nothing is known about the distribution of data. Did Authors analyze it? Was it parametric or non parametric? Which test was used? If the data was parametric the use of ANOVA is justified, but if it was non parametric the other tests should be used. All this doubts must be clarified.

Response 14. We used a completely randomized design, because the only source of variation was the additives, we did parametric analyzes and found other tests. “The data were processed using one-way analysis of variance (ANOVA), for the normality of the data the Kolmogorov Smirnov test was used and for the uniformity of the variance Bartlett's test was used, before carrying out the ANOVA. The Duncan’s test was used to determine the differences between means, according to the statistical software SPSS version 23 (SPSS. IBM -International Business Machine, Nueva York. USA) P values <0.05 were considered statistically significant.”

Point 15. One more important issue. The Authors did not provide any data on health status of pigs. In my opinion important and very useful would be the data about diarrhea occurrence. Looking at the table 2 data (average daily gains and feed conversion ratio) it seems to be clear that something was wrong in T0 group among day 42 and 49. So rapid decrease in gains must have a reason in health. However, in such a small group it also is possible, that the reason could be one sick pig. Such a pig should be excluded from analyses as outlier. That is why I would suggest to show SD or SEM for every group separately. Larger level of dispersion in group T0 would be important information. I would like the Authors also to clarify the n in consecutive analyses. I assume that weights and gains are individual, so n=12 per treatment. But it is not clear from the table. But what about feed intake and feed conversion ratio? I assume that n=3 (like the number of pens), but it is only assumption. This (n) should be clarified in every table, because there were important differences among analyses.

Response 15. The table was modified, we left the initial and final live weight for periods of 7 days and for the other variables (average daily gain, feed intake, and feed conversion ratio, kg / kg ) the analysis was done for the entire period.

Results

Point 16. Results are stated in 5 tables. In such a simple experimental design I would also calculate SEM separately for data inside groups (as I mentioned earlier). Sometimes such an analyses may show interesting, additional relations (e.g. if some parameters in one group are more dispersed than in the second one it is also important information). And last but not least, in every table n value for each group should be presented, because of imprecise description in M&M. The reader must know the number of animals in particular analyses to assess reliability and value of data.

Response 16. The table 2 and 5 were modified, and the discussion was changed.

Discussion

Point 17. The most of this section seem to be continued Introduction and Results in a little more expanded manner, rather than actual discussion. The most important information are only rewritten from results without any trial to explain or interpret. It is easy to say that the growth of pigs from T0 group was lower, but much more important is to try to find answer why. Such information should be analyzed with the trial to interpret. As I said earlier, maybe the reason was 1 or 2 pigs which experienced diarrhea. Interesting but confusing result is CSF vaccination effect. It cannot be stated that reaction was increased in T2 group, because the level of antibodies was almost the same before and after treatment. Important question is why the level of antibodies was increased in T2 in comparison to T1 and T0 before vaccination? And why, this level decreased in T0 and T1 group after vaccination? It does not make a sense. And the Authors did not try to analyze and interpret this data in Discussion.

Response 17. The discussion was rewrited.  In the  table 5 the information about CSF was deleted.

Point 18. To summarize, the manuscript needs substantial improvement, and must be completed with many important data and information. Without this improvement, it is very difficult to assess the reliability of presented data. The most important seem to be clarification of randomized model of division. Taking into account the character of data and experimental design, the number of 12 animals per group for production analyses, and 5 animals for lab analyses seem to be not enough for the full paper. I suggest major revision, but also to rewrite the manuscript as short communication.

Response 18. A clarification was made on this topic. We agree with the reviewers that this manuscript be made as a short communication.

Round 2

Reviewer 3 Report

No comments or suggestions

Author Response

Dear Reviewer,
Thank you very much for your review.

Reviewer 4 Report

General comment

I must say I am a little disappointed with the Authors answers, and even more disappointed with manuscript improvements. The manuscript still needs some minor language revision. I would suggest to look for some advise of native speaker in this point. However this is of minor importance. Much more important is that there are still many imprecision and unclear information, especially in M&M. Discussion also is almost completely not improved. The Authors have divided my review for 18 points with 18 answers. I will keep this division, to show which parts can be accepted, and which ones do not.

Specific comments

Response 1. The answer is acceptable and the improvement in manuscript also is acceptable

Response 2. The answer is acceptable and the improvement in manuscript also can be acceptable, however not perfect. There is still lack of information about some important issues concerning farm (floor type, ventilation system etc.)

Response 3. Acceptable

Response 4. Acceptable

Response 5. The answer does not match the question. I still cannot say if the replicates were provided the same time or not. It become a little more clear after response 9, but it is only the answer to review, without improvement in manuscript. It must be clarified in M&M. Line 103 weaned not weaning.

Response 6. I am a little confused. The question was about the concentration of pure active colistin in preparation, so that we can analyze the level of antibiotic intake per pig. The answer is 350 mg/1 kg of feed. In manuscript it is still 0,2 g/ 1 kg of feed. And it is still not clarified if those data are according to preparation or active substance. And which value is truth? Additional question. Which was the route of antibiotic administration? Was it added to diet or given separately as liquid by syringe, similarly to probiotic?

Response 7. This information is also a little confusing. Was it 5 mL or 10 mL? And if it was administered by syringe, why it is calculated per 1 kg of feed (line 106). It should be rather calculated per animal. Could you finally provide clear information about the dose of probiotic and antibiotic per one pig? One more question. The administration of probiotic using syringe is perfect method for scientific analyses because of precision, but it is impossible to use in commercial conditions. Do you have any idea how to use this preparation in commercial conditions? Should it be prepared in dried form as feed additive or in liquid form for administration via water? Some words about it in discussion would be advisable.  

Response 8. Not enough. There is still lack of information about crude fiber.

Response 9. The answer is acceptable, however, the information about the division of pen to 4 sections should be also stated in M&M of the manuscript. Especially that it also clarify doubts in point 5. Reading the manuscript without responses it is still unclear, and the Authors must remember that the reader will have no access to answers for review. Additional questions. In line 117 there is information about vaccination against ASF. To the best knowledge of reviewer there is no vaccine to ASF worldwide. In line 119 the information appeared about 12 repetitions. What does it mean?

Response 10. You offered 2 kg of food daily per pig, per pen (12 pigs), or per section in the pen (4 pigs)? This is not clarified nor in response and in M&M.

Response 11. Acceptable thanks to additional description in M&M line 127-128, (the information in response was unclear).

Response 12. The analyse was made daily or after each feeding?

Response 13. Such errors sometimes occurs, it is not a problem. However, it is still unclear why the Authors used 8 pigs for those analyses. Nine would be more sensible taking into account 3 replicates of experiment. I still would like to know the selection method and configuration of pigs selected to hematological analyses. Line 134, the Authors included 8 but did not erase 5.  

Response 14. Partially acceptable. The Authors focused on the last part of the point 14 concerning statistical analyses, but completely ignored the first part concerning the method to chose animals for experiment. They also completely omitted my doubts about the effect of much lower initial body weight of piglets in group T0. I accept information about statistical analyses, but I still expect clarification of my doubts in animal choosing. (I am not sure that completely randomized design is the best solution in such a small number of experimental animals. In my opinion, the animals should be selected to treatment groups by drawing 3 similar animals from one litter and divide them into 3 groups, 1 animal per group. In such a situation the design is still randomized, but the division of pigs from litters is more valuable for results. I would like the Authors to descript more detailed the system of randomized division. How many litters were used? Did they use every piglet from litters, or were they selected (only large or only middle weight pigs). It is especially important according to data in table 2, where the mean initial body weight in T0 is insignificantly, but in my opinion relevantly lower than in groups T1 i T2 (this two groups are almost identical). This difference can be pivotal for the rest of results in equal level to experimental factors).

Response 15. The answer does not match the question completely. I still expect the answer for point 15 and important improvements in manuscript taking into account every issue. (One more important issue. The Authors did not provide any data on health status of pigs. In my opinion important and very useful would be the data about diarrhea occurrence. Looking at the table 2 data (average daily gains and feed conversion ratio) it seems to be clear that something was wrong in T0 group among day 42 and 49. So rapid decrease in gains must have a reason in health. However, in such a small group it also is possible, that the reason could be one sick pig. Such a pig should be excluded from analyses as outlier. That is why I would suggest to show SD or SEM for every group separately. Larger level of dispersion in group T0 would be important information. I would like the Authors also to clarify the n in consecutive analyses. I assume that weights and gains are individual, so n=12 per treatment. But it is not clear from the table. But what about feed intake and feed conversion ratio? I assume that n=3 (like the number of pens), but it is only assumption. This (n) should be clarified in every table, because there were important differences among analyses).

Response 16. The answer does not match the question completely and there is no one modification in tables I have expected. (Results are stated in 5 tables. In such a simple experimental design I would also calculate SEM separately for data inside groups (as I mentioned earlier). Sometimes such an analyses may show interesting, additional relations (e.g. if some parameters in one group are more dispersed than in the second one it is also important information). And last but not least, in every table n value for each group should be presented, because of imprecise description in M&M. The reader must know the number of animals in particular analyses to assess reliability and value of data).

Response 17. The improvements of Discussion are insignificant. There is still no any trial to interpret data. It is only expanded description of results. No any information about the health of animals. Erasing the data about the effect of CSF vaccination is not enough to improve manuscript in terms of results and discussion. Especially looking at the conclusion. The Authors claim improving of immune function without affecting health indicators. Which health indicators were analysed? I have suggested to show the most important occurrence of diarrhea, but the Authors ignored this suggestion. I would like to see at least one health indicator based on disease symptoms analyses.  

To summarize, the manuscript still needs substantial improvement, and must be completed with many important data and information. I suggest another major revision, with more deep analyses of reviewer suggestions.

Author Response

General comment

I must say I am a little disappointed with the Authors answers, and even more disappointed with manuscript improvements. The manuscript still needs some minor language revision. I would suggest to look for some advise of native speaker in this point. However, this is of minor importance. Much more important is that there are still many imprecision and unclear information, especially in M&M. Discussion also is almost completely not improved. The Authors have divided my review for 18 points with 18 answers. I will keep this division, to show which parts can be accepted, and which ones do not.

Dear Reviewer 4,

Thanks for your letter and for the reviewers’ comments concerning our manuscript. Those comments are very valuable and helpful for revising our paper and guiding our research. We have studied those comments carefully and have made correction which include a change in the title of the manuscript.    Revised portion are marked colored in the paper.

The following is our point-by-point response to reviewers’ comments:

As general comments, this work used for the first time the bacterial strain registered in the GenBank “Lactobacillus plantarum CAM-6”; in in vitro experiments this strain was shown to have the optimal conditions to replicate and grow under different conditions, simulating the GIT of pigs (Betancur et al. 2020). As previously reported, we use a commercial diet normally sold for pig producers, we consider that the pigs were adequately fed according to the productive category under study and with the ingredients normally used in Colombia. In other words, this work could be replicated in any experimental area that used a feed that meets the nutritional requirements of the animals.

Our objective was to demonstrate that this strain was capable of modifying the growth performance in weaned pigs (being the most critical stage), the natural growth promoting effect of this bacterial strain similar to the subtherapeutic antibiotic is clearly observed and better than both treatments for the feed conversion ratio. We think this is the best result to visualize the importance of using this probiotic as an alternative to growth promoting antibiotics (GPA) in the pig industry, especially since 48 countries in the world still use GPA regularly. If these indicators do not increase with the use of the probiotic, it would be difficult to maximize the results to the industry. To the productive response, it is clearly associated with the immune response, it is known that IgA is part of the protection of the intestine and is increased with the probiotic, in addition the other indicators did not have significant effects, which shows that this probiotic does not cause apparent health complications. Our work was reviewed and corrected by a native scientist from the University of Arkansas, USA.

Specific comments

Point 2. The answer is acceptable and the improvement in manuscript also can be acceptable, however not perfect. There is still lack of information about some important issues concerning farm (floor type, ventilation system etc.).

Response 2. We add these words “The experimental area where the pigs were located is built with a concrete floor and precast concrete blocks for the fronts and sides of the corral. The pens are 0.6 m high, providing ample space for natural ventilation.”.

Point 5. The answer does not match the question. I still cannot say if the replicates were provided the same time or not. It become a little more clear after response 9, but it is only the answer to review, without improvement in manuscript. It must be clarified in M&M. Line 103 weaned not weaning.

Response 5. All replicates were used at the same time according to a completely randomized design because the only source of statistical variation was experimental treatments. We indicate it in the text. “The paragraph was modified “A total of 36 castrated piglets [(Landrace × Pietrain) × Duroc] weaned at 21 days were placed at the same time according to a random design with three experimental groups, 12 repetitions and 4 pigs per pen (3.60 x 1.90 each), where each corral formed an experimental unit. All animals were dewormed and vaccinated against classical swine fever at 42 days of age”

Point 6. I am a little confused. The question was about the concentration of pure active colistin in preparation, so that we can analyze the level of antibiotic intake per pig. The answer is 350 mg/1 kg of feed. In manuscript it is still 0.2 g/ 1 kg of feed. And it is still not clarified if those data are according to preparation or active substance. And which value is truth? Additional question. Which was the route of antibiotic administration? Was it added to diet or given separately as liquid by syringe, similarly to probiotic?

Response 6. We use 350 mg/kg feed of colistin sulfate (20% of the active product) as a growth promoter on pig diets. This was clarified in the text.

Point 7. This information is also a little confusing. Was it 5 mL or 10 mL? And if it was administered by syringe, why it is calculated per 1 kg of feed (line 106). It should be rather calculated per animal. Could you finally provide clear information about the dose of probiotic and antibiotic per one pig? One more question. The administration of probiotic using syringe is perfect method for scientific analyses because of precision, but it is impossible to use in commercial conditions. Do you have any idea how to use this preparation in commercial conditions? Should it be prepared in dried form as feed additive or in liquid form for administration via water? Some words about it in discussion would be advisable.  

Response 7. Thank you very much for clarifying the doubt of the possible readers. We use the probiotic orally, not with feed. That is, a basal diet was supplied without antibiotics or additives, the probiotic was administered orally through a syringe. We use 5 ml of the probiotic biopreparation per pig per day. The piglets were included in the study from day 21 to day 49 post-weaning.  This statement has been added to the text.  Furthermore, you are right, further studies need to be conducted to evaluate the efficacy of this strain under commercial conditions and water administration.  This statement has been included in the abstract and discussion.  Hence, we also change the scope and title of the manuscript, thank you.

Point 8. Not enough. There is still lack of information about crude fiber.

Response 8. We add the crude fiber information.  

Point 9. The answer is acceptable, however, the information about the division of pen to 4 sections should be also stated in M&M of the manuscript. Especially that it also clarifies doubts in point 5. Reading the manuscript without responses it is still unclear, and the Authors must remember that the reader will have no access to answers for review. Additional questions. In line 117 there is information about vaccination against ASF. To the best knowledge of reviewer there is no vaccine to ASF worldwide. In line 119 the information appeared about 12 repetitions. What does it mean?

Response 9. Thank you very much for your comments. We corrected the number of replicates and deleted the vaccination against African swine fever, there was a confusion with the classical swine fever. We add these words “The trial (28 days) used 36 piglets weaned at 21 days of age that were randomly divided into three experimental groups under the same productive conditions, with three replicates and four pigs per pen (3.60 x 1.90 each), where each pen was an experimental unit. All animals were dewormed and vaccinated against classical swine fever at 42 days of age”

Point 10. You offered 2 kg of food daily per pig, per pen (12 pigs), or per section in the pen (4 pigs)? This is not clarified nor in response and in M&M.

Response 10. We add these words “In each pen, 2 kg of feed per pig was supplied at two frequencies (8:00 am and 3:00 pm) in linear canoe-type feeders. The water was supplied at will in metal nipple type waterers.”

Point 12. The analyse was made daily or after each feeding?

Response 12. We add these words During the study, live weights of the pigs were determined weekly from 21 to 49 days (4 weeks) using a scale with a precision ±1 g (Mettler Toledo, Digital Scale, Ohio, USA), at the same time of day and before feeding. Feed intake was determined daily, and it was accumulated by experimental week, the difference between feed supply and rejection was considered. The feed conversion rate was calculated as the amount of ingested feed required to gain 1 kg of LW. The average daily LW gain was determined from the difference between the final and the initial LW during the study”.

Point 13. Such errors sometimes occurs, it is not a problem. However, it is still unclear why the Authors used 8 pigs for those analyses. Nine would be more sensible taking into account 3 replicates of experiment. I still would like to know the selection method and configuration of pigs selected to hematological analyses. Line 134, the Authors included 8 but did not erase 5.  

Response 13. Thank you very much for your question. For the hematological study we selected 8 pigs (66.66% of the animals per group) at random for each treatment, we considered an adequate number to find significant differences for these parameters. We add these words “eight pigs from each treatment group were randomly selected and 5 mL of blood were collected from their jugular veins with 21 G needles”.

Point 14. Partially acceptable. The Authors focused on the last part of the point 14 concerning statistical analyses, but completely ignored the first part concerning the method to chose animals for experiment. They also completely omitted my doubts about the effect of much lower initial body weight of piglets in group T0. I accept information about statistical analyses, but I still expect clarification of my doubts in animal choosing. (I am not sure that completely randomized design is the best solution in such a small number of experimental animals. In my opinion, the animals should be selected to treatment groups by drawing 3 similar animals from one litter and divide them into 3 groups, 1 animal per group. In such a situation the design is still randomized, but the division of pigs from litters is more valuable for results. I would like the Authors to descript more detailed the system of randomized division. How many litters were used? Did they use every piglet from litters, or were they selected (only large or only middle weight pigs). It is especially important according to data in table 2, where the mean initial body weight in T0 is insignificantly, but in my opinion relevantly lower than in groups T1 i T2 (this two groups are almost identical). This difference can be pivotal for the rest of results in equal level to experimental factors).

Response 14. We only had one source of statistical variation (the experimental treatments), so we used a completely randomized design. Pigs at 21 days were randomly selected at the same time from 20 sows that gave birth on the same day, it is clearly observed that the initial live weight did not show statistical differences, so this indicator did not influenced subsequent weights, therefore a covariance was not used. The experimental design is determined according to the sources of statistical variation, not by the N, although the Latin square, used in digestibility or energy studies (for example), uses few animals, which was not the case in these growing animals. In summary, we randomly select pigs from 20 sows on the same day of birth and weaned. We add these words “Piglets born and weaned at the same time (21 days old) from 20 sows apparently healthy Yorkshire x Landrace (three farrowing’s) located in the farrowing area were randomly selected; farrowing’s were synchronized and no abnormalities were found during pregnancy and lactation.  Statistical section has been corrected, thank you.

Point 15. The answer does not match the question completely. I still expect the answer for point 15 and important improvements in manuscript considering every issue. (One more important issue. The Authors did not provide any data on health status of pigs. In my opinion important and very useful would be the data about diarrhea occurrence. Looking at the table 2 data (average daily gains and feed conversion ratio) it seems to be clear that something was wrong in T0 group among day 42 and 49. So rapid decrease in gains must have a reason in health. However, in such a small group it also is possible, that the reason could be one sick pig. Such a pig should be excluded from analyses as outlier. That is why I would suggest to show SD or SEM for every group separately. Larger level of dispersion in group T0 would be important information. I would like the Authors also to clarify the n in consecutive analyses. I assume that weights and gains are individual, so n=12 per treatment. But it is not clear from the table. But what about feed intake and feed conversion ratio? I assume that n=3 (like the number of pens), but it is only assumption. This (n) should be clarified in every table, because there were important differences among analyses).

Response 15. It is important to highlight that apparently healthy animals were selected and during the experimental period the animals did not present any signs, symptoms or other pathologies associated with any disease. The diarrheal syndrome was also not found, which shows that the conditions of tenure were adequate. Clearly from 35 days of life, the antibiotic and probiotic increased live weight, which shows its growth promoting effect and the justification for its use. In addition, when the weight gain of the treatments from 42 to 49 days is compared, it is observed that the T0 = 1.71 kg, T1= 1.63 kg and T2 = 2.08 kg, clearly shows that the T0 for this stage grew more than the antibiotic group, this shows that the growth was according to the productive response. It is important to remember that the basal diet has neither antibiotics nor additives. We clarify in the table the N for each case.

DS and CV are known to be used for descriptive statistics, however when comparing treatments, it is correct to use the SEM. The standard error of the mean (SEM), is the error due to the estimation of the population mean from the sample means: , this means that the The SEM will be unique for each media when the number of observations per treatment is different, and this was not the case in our study, which had the same number of observations for each treatment, that's why we use a single SEM for all means. However, we inform you that the SD was less than 0.60. Also, as reported in the manuscript for the normality of the data the Kolmogorov Smirnov test was used and for the uniformity of the variance Bartlett's test was used, before carrying out the ANOVA, this clearly shows the data were not scattered and allowed to find significant differences among treatments.

Point 16. The answer does not match the question completely and there is no one modification in tables I have expected. (Results are stated in 5 tables. In such a simple experimental design I would also calculate SEM separately for data inside groups (as I mentioned earlier). Sometimes such an analyses may show interesting, additional relations (e.g. if some parameters in one group are more dispersed than in the second one it is also important information). And last but not least, in every table n value for each group should be presented, because of imprecise description in M&M. The reader must know the number of animals in particular analyses to assess reliability and value of data).

Response 16. As mentioned above, a separate SEM cannot be indicated for each mean, because all the treatments had the same N, in addition the SEMs are very low, allowing significant differences to be found. Clearly, the data was not dispersed, because found differences, and the SEM was low. We add in text the N for each case.

Point 17. The improvements of Discussion are insignificant. There is still no any trial to interpret data. It is only expanded description of results. No any information about the health of animals. Erasing the data about the effect of CSF vaccination is not enough to improve manuscript in terms of results and discussion. Especially looking at the conclusion. The Authors claim improving of immune function without affecting health indicators. Which health indicators were analysed? I have suggested to show the most important occurrence of diarrhea, but the Authors ignored this suggestion. I would like to see at least one health indicator based on disease symptoms analyses.  

To summarize, the manuscript still needs substantial improvement, and must be completed with many important data and information. I suggest another major revision, with more deep analyses of reviewer suggestions.

For answered responses 14, 15, 16 and 17,

Response 17. Hematological parameters are considered indicators of health in animals and humans, in this case the most important indicators were within normal parameters. In general, when diarrheal syndrome occurs, it causes a hemoconcentration of the blood with an increase in hematocrit, and this was not the case. On the other hand, it is important to emphasize that diarrheal syndrome is not a disease, it is a syndrome, that is, symptoms and the same signs with different etiologies, in weaned pigs, diarrhea occurs due to multifactorial factors, the most frequent being the change in diet and separation from mothers, in this case diarrhea is not synonymous with disease, except that it presents some symptoms of some disease. We repeat again, that the SEM was below what allowed with the used repetitions to find significant differences, this clearly shows the little dispersion of the data. We add some paragraphs in the discussion.
